# Development and Characterization of Environmentally Friendly Wood Plastic Composites from Biobased Polyethylene and Short Natural Fibers Processed by Injection Moulding

**DOI:** 10.3390/polym13111692

**Published:** 2021-05-22

**Authors:** Celia Dolza, Eduardo Fages, Eloi Gonga, Jaume Gomez-Caturla, Rafael Balart, Luis Quiles-Carrillo

**Affiliations:** 1Textile Industry Research Association (AITEX), Plaza Emilio Sala, 1, 03801 Alcoy, Spain; cdolza@aitex.es (C.D.); efages@aitex.es (E.F.); egonga@aitex.es (E.G.); 2Technological Institute of Materials (ITM), Universitat Politècnica de València (UPV), Plaza Ferrándiz y Carbonell 1, 03801 Alcoy, Spain; rbalart@mcm.upv.es (R.B.); luiquic1@epsa.upv.es (L.Q.-C.)

**Keywords:** BioHDPE, environmentally friendly composites, short natural fibers, added-value green composites, mechanical properties

## Abstract

Environmentally friendly wood plastic composites (WPC) with biobased high density polyethylene (BioHDPE) as the polymer matrix and hemp, flax and jute short fibers as natural reinforcements, were melt-compounded using twin-screw extrusion and shaped into pieces by injection molding. Polyethylene-graft-maleic anhydride (PE-g-MA) was added at two parts per hundred resin to the WPC during the extrusion process in order to reduce the lack in compatibility between the lignocellulosic fibers and the non-polar polymer matrix. The results revealed a remarkable improvement of the mechanical properties with the combination of natural fibers, along with PE-g-MA, highly improved stiffness and mechanical properties of neat BioHDPE. Particularly, hemp fiber drastically increased the Young’s modulus and impact strength of BioHDPE. Thermal analysis revealed a slight improvement in thermal stability with the addition of the three lignocellulosic fibers, increasing both melting and degradation temperatures. The incorporation of the fibers also increased water absorption due to their lignocellulosic nature, which drastically improved the polarity of the composite. Finally, fire behavior properties were also improved in terms of flame duration, thanks to the ability of the fibers to form char protective barriers that isolate the material from oxygen and volatiles.

## 1. Introduction

In recent years, social awareness has undergone a significant rise, due to environmental issues related to waste generation, petroleum shortage and an increasing need for reducing the carbon footprint. The global plastic production now stands at around 300 Mt/year; thus confirming the great amount of residues generated by the plastic industry [1]. Those trends have made plastic industry to turn to more environmentally friendly polymeric materials and composites [2]. Obtaining new materials from renewable resources or even from wastes, has gained important attention [3,4]. The increasing use of biopolymers (biobased or biodegradable petroleum-derived biopolymers) has revealed a clear contribution to diminishing the carbon footprint, in comparison with non-biodegradable petroleum-derived polymers plastics [5,6].

Biobased polymers can successfully diminish the use of fossil resources through the use of biomass or renewable resources, thus reducing the carbon footprint [7,8]. Within the plastic industry, high density polyethylene (HDPE) is one of the most globally used commercial plastics, just after polyvinyl chloride (PVC), and polypropylene (PP) in terms of worldwide production and use [9]. It is for this reason that biobased polyethylene (BioHDPE), also known as green polyethylene, is a good solution for reducing the problems derived from the use of fossil resources. This green polyolefin is produced by ethylene polymerization, obtained from catalytic dehydration of bioethanol [10]. BioHDPE has the same properties as its petrochemical counterpart, which possesses good mechanical resistance, high ductility and improved water resistance [4,11]. In 2018, biobased polyethylene represented approximately 9.5% of the global biopolymer production capacity, reaching a production annual rate close to 200,000 t/year [12]. Normally, injection-molded BioHDPE pieces can be used for fabrication of plastic parts and components, as well as packaging and flexible films [11].

Together with the use of new polymers from renewable resources, the interest in natural fiber reinforced plastics (NFRP) [13,14,15] and wood plastic composites (WPC) [16,17,18] has remarkably increased. Since most of these natural fibers or particles can be obtained from waste biomass, agroforestry or industrial wastes, this kind of lignocellulosic reinforcements can positively contribute to obtaining high environmentally friendly composites [19,20,21]. These materials can also be directly focused on reducing environmental limitations underlying in a linear economy. Thus allowing to reutilize and upgrade different industrial wastes, thus contributing to a transition from a linear economy to a circular economy (CE) [22]. This change from the waste removal concept to the conversion of wastes into high added-value materials is the core of the transition towards a circular economy [23]. In the last years, natural fillers and wastes have been widely used as fillers/reinforcements in polymer composites [24,25]. Those fillers can be derived from minerals, animals or plants. However, plant fillers, either in particle or fiber form, are the ones that can be found more frequently in green composites [26,27]. This is because they can be obtained from agricultural and industrial wastes, or by-products from food processing, which allows to improve the valorization of discarded materials and promote circular economy [28,29]. In this sense, recently, new materials composed of biosourced wastes have been developed, including particle-filler polymers with orange peel flour [30], babasu shell flour [31], almond shell flour [32], or fibers such as pineapple leaf fiber [33], jute fiber [34], banana pseudostem fiber [35], coconut fiber [36], flax fiber [37], etc.

In particular, it is plant fibers which may have an important role in developing biodegradable composites to overcome the current environmental issues in composites, especially related to the recyclability problems of glass fiber-reinforced plastics (GFRP). In addition, the use of natural fibers from waste textile is a way to give a second use to these wastes. In particular, natural fibers such as hemp fiber [38,39], flax [40,41], jute [42], sisal [43], abutilon fiber [44] or *Arundo Donax* L. fiber [45] are particularly attractive in applications such as automotive or transport because of lower cost and lower density composites are emerging as realistic alternatives to glass-reinforced composites in many applications [46]. Moreover, the main waste from some food industries and agroforestry are lignocellulosic materials, which have the potential to provide reinforcing properties to polymers due to their availability, low weight, renewable nature, biodegradability and low cost. However, each type of plant has a specific chemical composition, which makes fibers from each plant present different chemical compositions on their own. Furthermore, there can be slight differences in chemical composition in terms of cellulose, lignin and hemicellulose contents in fibers originated from the same plant, depending on the part of the plant that they come from [47]. Therefore, due to this microstructural variability, vegetal fibers show some disadvantages and, in some applications, they do not meet the expected reinforcing effect on polymeric matrices. Furthermore, their reinforcing behavior shows some uncertainty under short and long time loads [48]. Therefore, a comparison of the features of different natural fibers in the same polymeric matrix, becomes an interesting topic.

Flax is one of the most used plants in industries for decades. The worldwide production was approximately 3 Mt in 2017, Canada, China, India and United States being the main producers [49]. This fiber stands out for its high cellulose content (~78%) [50], and the disorientation of its cellulose microfibers [51]. Those features provide flax with excellent mechanical response in comparison to other vegetable fibers, and makes flax fiber (FF) an interesting candidate as reinforcement in polymers. On the other hand, hemp fiber is one of the cheapest natural fibers and can be easily obtained. For this reason, hemp has attracted great attention of researchers and producers from Europe and North America. Hemp contains approximately 61% cellulose, 24% hemicellulose, 10% lignin and 3% extracts [52]. Several researchers have taken advantage of the reinforcing potential of hemp fibers in order to develop thermoplastic and thermosetting composites using different processing techniques [53,54]. Finally, jute fibers are gaining increased attention as lignocellulosic fibers. Jute is a well-known cellulosic reinforcement and is easily available in India and Bangladesh at low cost. It contains cellulose (45–72%), hemicellulose (13.6–21%), lignin (12–26%), and pectin 0.2% [55,56].

However, the incorporation of vegetable fibers (highly polar elements) into a polymer matrix (normally non-polar) represents a challenge, because their difference in polar behavior makes them incompatible [57]. This lack of compatibility is caused by the difficulty of both elements to form chemical bonds between them, which usually produces a lack of adhesion between the embedded fibers and the surrounding polymeric matrix which, in turn, leads to poor mechanical properties. Therefore, in order to obtain good mechanical properties, this compatibility issue must be solved and high fiber–polymer interactions are required [58]. Copolymers can positively contribute to this end. They present a dual functionality that can interact with both polymer matrix and natural fiber reinforcement. One of the most used copolymers in natural fiber reinforced plastics is polyethylene-*graft*-maleic anhydride (PE-*g*-MA). The dual functionality of this copolymer allows interactions between the hydrophobic polymer and the PE-chains in PE-*g*-MA, while the grafted maleic anhydride (MA) can interact, or even react, with the hydroxyl groups in cellulose, thus leading to a compatibilization effect. This compatibilizer has given promising results in composites of recycled polyethylene with poplar, improving their mechanical properties and reducing their water absorption capacity [59]. However, it is yet to be thoroughly tested with particular fiber composites, such as jute and flax fiber composites.

In this work, a series of environmentally friendly wood plastic composites have been obtained by injection moulding and subsequently, characterized. It is important to bear in mind that lignocellulosic materials degrade at moderate temperatures; therefore, the selected polymer matrix should be processed at temperatures below 200–220 °C, since hemicellulose degrades above these temperatures. Biobased polyethylene (BioHDPE) offers interesting environmental issues since it is obtained from renewable sources. Nevertheless, its price it still higher than its corresponding petroleum-based counterpart. Therefore, manufacturing of wood plastic composites with BioHDPE represents an interesting solution to obtain price competitive materials with wood-like appearance. In this work, we propose the use of different short vegetable fibers as reinforcing fillers, namely, hemp, flax and jute fibers. PE-*g*-MA is used as compatibilizer to improve fiber–polymer interactions. The objective of developing such materials is to provide high added-value wood-like materials, by using BioHDPE-based as high environmental efficiency polymer matrix. In addition, the aim is to compare the different fibers in order to find the one that provides the best balanced properties. The mechanical, morphological, thermal, and thermomechanical properties of the injection-molded green composites have been studied with the aim of determining which fiber can provide the best balanced properties to biobased polyethylene matrix. In order to compare the properties between the materials with different fibers, the same fiber/BioHDPE ratios have been chosen for each of the fibers. Although this could prove to be another research limitation, as no other ratios are tested, these ratios have been selected according to the work of Arrakhiz et al. [60], who reported that a 20 wt% ratio of natural fiber in polyethylene composites showed the best mechanical and thermal properties in comparison with other ratios.

## 2. Materials and Methods

### 2.1. Materials

Biobased high density polyethylene (BioHDPE), SHA7260 grade, was supplied in pellet form by FKuR Kunststoff GmbH (Willich, Germany) and manufactured by Braskem (São Paulo, Brazil). This green polyethylene has a density of 0.955 g/cm^3^ and a melt flow index (MFI) of 20 g/10 min, measured with a load of 2.16 kg and a temperature of 190 °C.

A polyethylene-graft-maleic anhydride (PE-*g*-MA) copolymer with CAS Number 9006–26–2 and MFI values of 5 g/10 min (190 °C/2.16 kg), was obtained from Sigma-Aldrich S.A. (Madrid, Spain). The proportion of maleic anhydride (MA) in the copolymer is 0.5 wt% according to the supplier. This PE-based copolymer was selected due to its dual functionality (polar/non-polar).

Three different short natural fibers were used in this study to give green composites with biobased polyethylene, namely, hemp, flax and jute fibers. All fibers were supplied by SCHWARZWÄLDER TEXTIL-WERKE (Schenkenzell, Germany). Hemp fibers showed an irregular rough shape with an average diameter of 15–50 μm, specific gravity 1.48–1.50 g/cm^3^ and elongation at break of 1.3%. Flax fiber had a polygonal shape with a thin oval lumen. The fiber diameter varied in the 10–500 μm range with a density of 1.4–1.5 g/ cm^3^, and an elongation at break of 1.2–1.5%. Finally, jute fiber has a polygonal (multi cornered) cross-section with a fiber diameter comprised between 30 and 500 μm, specific gravity 1.44 g/ cm^3^ and elongation at break of 1.3%. Figure 1 shows the visual appearance of the fibers and their shape at a magnification of 50×.

### 2.2. Sample Preparation

BioDHPE and fibers were dried at 60 °C for 48 h in a dehumidifying dryer MDEO (Industrial Marsé, Barcelona, Spain) to remove any residual moisture prior to processing. The materials were then fed into the main hopper of a co-rotating twin-screw extruder from Construcciones Mecánicas Dupra, S.L. (Alicante, Spain). This extruder has a screw diameter of 25 mm with a length-to-diameter ratio (L/D) of 24. The extrusion process was carried out at 20 rpm, setting the temperature profile, from the hopper to the die, as follows: 140–145–150–155 °C. The different BioHDPE composites were extruded through a round die to produce strands and, subsequently, pelletized using an air-knife unit. In all cases, residence time was approximately 1 min. Table 1 summarizes the compositions and coding.

The compounded pellets were, thereafter, shaped into standard samples by injection molding in a Meteor 270/75 from Mateu & Solé (Barcelona, Spain). The temperature profile in the injection molding unit was 140 °C (hopper), 150 °C, 155 °C, and 160 °C (injection nozzle). A clamping force of 75 tons was applied while the cavity filling and cooling times were set to 1 and 10 s, respectively. Standard samples for mechanical and thermal characterization with an average thickness of 4 mm were obtained.

### 2.3. Material Characterization

#### 2.3.1. Mechanical Tests 

In order to obtain the mechanical properties, tensile tests were carried out in a universal testing machine ELIB 50 from S.A.E. Ibertest (Madrid, Spain) using injection-molded dog bone-shaped samples as indicated by ISO 527–1:2012. A 5-kN load cell was used and the cross-head speed was set to 20 mm/min. Shore hardness was measured in a 676-D durometer from J. Bot Instruments (Barcelona, Spain), using the D-scale, on injection-molded samples with dimensions 80 × 10 × 4 mm^3^, according to ISO 868:2003. Toughness was also studied on injection-molded rectangular samples with dimensions of 80 × 10 × 4 mm^3^ by the Charpy impact test with a 1-J pendulum from Metrotec S.A. (San Sebastián, Spain) on notched samples (0.25 mm radius v-notch), following the specifications of ISO 179–1:2010. All tests were performed at room temperature, that is, 25 °C, and at least 6 samples of each material were tested, and their values averaged.

#### 2.3.2. Morphology

The morphology of the fracture surfaces of the BioHDPE-natural fiber composites, obtained from the impact tests, was observed by field emission scanning electron microscopy (FESEM) in a ZEISS ULTRA 55 microscope from Oxford Instruments (Abingdon, UK), working at an acceleration voltage of 2 kV. Before placing the samples in the vacuum chamber, they were sputtered with a gold–palladium alloy in an EMITECH sputter coating SC7620 model from Quorum Technologies, Ltd. (East Sussex, UK).

#### 2.3.3. Thermal Analysis

The main thermal transitions of BioHDPE-natural fiber composites were obtained by differential scanning calorimetry (DSC) in a Mettler-Toledo 821 calorimeter (Schwerzenbach, Switzerland). An average sample weight ranging from 5 to 7 mg was subjected to the following three-step dynamic thermal cycle: first heating from 20 °C to 160 °C followed by cooling to 0 °C, and second heating to 250 °C. Heating and cooling rates were set to 10 °C/min. All tests were run in nitrogen atmosphere with a flow rate of 66 mL/min using standard sealed aluminum crucibles (40 μL). The degree of crystallinity ($\chi_{c}$) was determined following the Equation (1):(1)$$\chi_{c}\left(\%\right)=\left[\frac{\Delta H_{m}}{\Delta H_{m}^{0}\cdot \left(1-w\right)}\right]\cdot 100$$
where ΔH_m_ (J/g) stands for the melting enthalpy of the sample, Δ$H_{m}^{0}$ (J/g) represents the theoretical melting enthalpy of a fully crystalline BioHDPE, that is, 293.0 J/g [61], and w corresponds to the weight fraction of different fibers in the formulation.

Thermogravimetric analysis (TGA) was performed in a LINSEIS TGA 1000 (Selb, Germany). Samples with an average weight between 5 and 7.5 mg were placed in standard alumina crucibles of 70 µL and subjected to a heating program from 30 °C to 700 °C at a heating rate of 10 °C/min in air atmosphere. The first derivative thermogravimetric curves (DTG) were also determined, expressing the weight loss rate as the function of time. All tests were carried out in triplicate.

#### 2.3.4. Thermomechanical Characterization

Dynamical mechanical thermal analysis (DMTA) was carried out in a DMA1 dynamic analyzer from Mettler-Toledo (Schwerzenbach, Switzerland), working in single cantilever flexural conditions. Injection-molded samples with dimensions of 20 × 6 × 2.7 mm^3^ were subjected to a dynamic temperature sweep from −160 °C to 100 °C at a constant heating rate of 2 °C/min, a frequency of 1 Hz and a maximum cantilever deflection of 10 µm.

#### 2.3.5. Color Measurements

A Konica CM-3600d Colorflex-DIFF2 colorimeter, from Hunter Associates Laboratory, Inc. (Reston, VA, USA) was used for the color measurement. Color indexes (L*, a*, and b*) were measured according to the following criteria: L * is the lightness and changes from 0 to 100; a* stands for the green (a* < 0) to red (a* > 0) color coordinate, while b*, represents the blue (b* < 0) to yellow (b* > 0) color coordinate. Measurements were done in triplicate.

#### 2.3.6. Water Uptake Characterization

Injection-molded samples of 4 × 10 × 80 mm^3^ were immersed in distilled water at 23 ± 1 °C. The samples were taken out and weighed weekly using an analytical balance with a precision of ± 0.1 mg, after removing the residual water with a dry cloth. The evolution of the water absorption was followed for a period of 15 weeks. Measurements were performed in triplicate.

#### 2.3.7. Fire Behavior

In order to obtain the fire behavior of the BioHDPE-natural fiber composites, calorimetry and opacity tests were carried out. The equipment used for calorimetry characterization was a PARR 6200 oxygen bomb calorimeter (Parr Instrument Company, Moline, IL, USA). The samples in pellet form were turned into fine powder by a grinder and liquid nitrogen was used in order to avoid thermal decomposition. The temperature of the distilled water was set at 26 °C and the closing pressure was set between 3.0 and 3.5 MPa with no air in the inside. Finally, in order to obtain the smoke density, the opacity test was carried out in a chamber model NBS Smoke Chamber (Concept Equipment Ltd., East Sussex, UK). The dimensions of the samples were 75 × 75 × 5 mm^3^. The samples must be conditioned to a constant mass at a temperature of 23 °C and a relative humidity of 50% according to ISO 291. The samples were wrapped in aluminum foil (0.04 mm thick) and exposed to a radiation of 50 kW/m^2^, 25 mm conical distance and 600 s test time. 

Regarding the calculation of the specific optical density (D_s_), Equation (2) was used:(2)$$D_{s}\left(t\right)=\frac{V}{\mathrm{AL}}\log_{10}\frac{100}{T\left(t\right)}\left[\mathrm{Adimensional}\right]$$
where D_s_(t) is the specific optical density; $\frac{V}{\mathrm{AL}}$ is the ratio between the volume of the camera (V), the exposed area of the specimen (A) and the length of the light path (L). This ratio is equivalent to 132 and, finally, T(t) is the value of transmittance measured in %.

## 3. Results

### 3.1. Mechanical Properties of BioHDPE-Natural Fiber Composites

Mechanical characterization of the injection-molded wood plastic composites of BioHDPE and different natural fibers provides relevant information regarding properties and possible applications of these materials. Mechanical properties obtained through the carried out tests are shown in Table 2. In particular, the tensile modulus(E), the maximum tensile strength (σ_max_) and, the elongation at break (ε_b_) of BioHDPE-natural fiber composites with hemp, flax and jute fibers, compatibilized with PE-g-MA can be observed. Additionally, Figure 2 shows the stress–strain curves for all the BioHDPE composites reinforced with fibers and compatibilized with PE-g-MA.

It can be directly seen how the neat BioHDPE showed typical values for E (0.790 GPa) and σ_max_ (14.5 MPa), and as many other HDPE the elongation at break is extremely high; in fact, the material did not break during the test at a cross-head speed of 5 mm/min, as it can be verified in Figure 2. These values are indicative of a material with moderate-to-low stiffness, but great ductility above all. Similar values were reported by other authors for the same biobased HDPE [4]. The addition of PE-*g*-MA into the BioHDPE matrix, does not change in a remarkable way the overall properties. Despite there not being a direct change in tensile properties, the presence of PE-*g*-MA seems to enhance some plasticization of BioHDPE, which can be observed in the morphology section (see Figure 3), where a higher plastic deformation can be assessed from FESEM observation. Lima et al. [62] showed how the incorporation of 5% and 10% of PE-*g*-MA into a HDPE matrix did not change the maximum strength and Young’s modulus, while an increase in the elongation at break was observed. In addition to this, the most important role of PE-*g*-MA in BioHDPE-natural fiber composites is compatibilization due to its dual functionality. Effah et al. [63] showed an increase in the Young’s modulus in WPC of low density polyethylene (LDPE) and different woods, by using PE-*g*-MA as compatibilizer.

The incorporation of different vegetable fibers into the BioHDPE/PE-g-MA blend, provides very relevant data for the industrial application of these materials. The addition of 20 wt.% hemp fibers increases the modulus up to 1.730 GPa, which reveals a 118% increase with regard to neat BioHDPE. The addition of a non-reinforcing filler into a polymer matrix, usually leads to a decreased tensile strength due to poor polymer-filler interaction. Nevertheless, the maximum tensile strength for BioHDPE composites with hemp fiber, does not decrease. In fact, it increases slightly to 15.5 MPa, thus verifying the reinforcing effect provided by hemp fibers. Yomeni et al. [64] reported similar results for composites of LDPE and 30% of treated hemp fiber, obtaining an increase in both the modulus and resistance of 151% and a 36%, respectively. As expected, the matrix continuity is reduced by the presence of the embedded hemp fibers which, in turn, decreases the overall cohesion and, therefore, the elongation at break is reduced down to 3.4%. Mazzanti et al. [65], observed a great reduction in the elongation at break in PLA-composites with hemp fiber, even with lower fiber content. The increase in the tensile modulus is typical of polymer-filled materials since the tensile modulus stands for the applied stress and the elongation in the linear/elastic region. Due to the dramatic decrease in elongation at break, the increase in tensile modulus is evident. Nevertheless, the good tensile strength of the BioHDPE composite with hemp fiber, which reveals a clear reinforcing effect, is worth noting.

With regard to BioHDPE composites with flax fiber, the tensile properties are similar to those obtained in composites with hemp fiber. The Young’s modulus increases up to 1.630 GPa, and the maximum tensile strength increases even further up to 16.7 MPa. In a similar way, the elongation at break is dramatically reduced down to 2.8%. This decrease could be due to the nature, length, and size of the flax fiber, which is the smallest of the three natural fibers used in this work (Figure 1b). Accordingly to these results, Zhang et al. [66], reported similar reinforcing properties in HDPE-flax composites. More specifically, they observed that above 12% flax fiber, the reinforcing effect was lost.

Finally, the incorporation of jute fibers into the BioHDPE matrix, also leads to interesting tensile properties. While the tensile modulus is similar to that obtained in flax and hemp composites, BioHDPE-jute fiber composites exhibits the highest tensile strength and elongation at break from all the studied formulations, with a value of 18.6 MPa and 4.4%, respectively. Those values are highly related to the size and shape of the fibers (Figure 1c). It must be remarked that jute fiber is the one with the highest elongation at break. Agüero et al. [37] showed how the incorporation of short flax fibers into polylactide (PLA), worsens all mechanical properties if a compatibilizing agent is not included.

In general, the addition of natural fibers into BioHDPE provides higher hardness, as it can be seen in Table 2. Initially, it can be seen how the addition of PE-*g*-MA slightly reduces hardness due to some plasticization effect as above-mentioned (see Figure 3). On the other hand, the addition of different fibers increases Shore D hardness from 56.0 (BioHDPE) to 59.4, 61.6 and 60.2 for hemp, flax and jute composites, respectively. This increase is directly related to the intrinsic hardness of the lignocellulosic fibers, being flax fiber the one with the highest hardness. Regarding impact strength, the values obtained report on very promising results in terms of some technical applications. Neat BioHDPE is a very ductile material with a relatively high impact strength (2.7 kJ/m^2^), even in notched samples. It should be noted that impact strength is directly related to two mechanical parameters, namely the supported stress and the deformation before break. The addition of all three fibers into the BioHDPE matrix, provides a clear increase in impact energy absorption, with values ranging between 3 and 4 kJ/m^2^. Particularly, the use of hemp as reinforcing fiber improves the impact strength by 44% with respect to neat BioHDPE. This improvement is closely linked with the ability of these fibers to transfer loads longitudinally. This is based on the fracture resistance theory. When these BioHDPE composites are subjected to impact conditions, numerous micro cracks are produced at the very early stages of the impact. Then, the fibers stretch along these micro cracks, thus stopping their growth. As a consequence of this, the presence of fibers significantly improves impact properties as observed in polyvinyl chloride (PVC)-based WPC [67]. As it has been previously mentioned, due to flax fibers being shorter, BioHDPE composites with flax fibers offer the least impact absorbed energy.

### 3.2. Morphology of BioHDPE-Natural Fiber Composites

Figure 3 shows the difference in fracture morphology when PE-*g*-MA is added to the BioHDPE matrix, with a magnification of 2000×. The addition of PE-*g*-MA (Figure 2) promotes a clear increase in rough surface, which directly improves the ductility of the blend. This morphology corroborates previous findings regarding the effect of PE-*g*-MA on mechanical properties. In this case, the improvement in ductility due to PE-*g*-MA addition can be proved by FESEM observation. These results are closely related to the results reported by Lima et al. [62], where PE-*g*-MA provided an improvement in ductility and plasticization of a biobased high density polyethylene and its compounds. 

Figure 4 shows the FESEM images corresponding to the fracture surface of BioHDPE-natural fiber composites, compatibilized with PE-*g*-MA. Figure 4a shows the morphology and fiber distribution in BioHDPE-hemp composites. As it can be seen, hemp fibers have a diameter between 20 and 60 µm and offer good interaction with the polymer, which can be assessed by the small gap between the fiber and the surrounding matrix, which has a very positive effect on load transfer and, subsequently, improved toughness. On the other hand, Figure 4b shows the fracture morphology corresponding to BioHDPE-flax composites. It can be seen that flax fibers are the thinnest fibers of the three studied ones. Moreover, some spherical particles can be appreciated, which could be responsible for the low impact strength and elongation observed in mechanical characterization. It can be concluded from FESEM images that PE-*g*-MA enhances good polymer-flax interactions since an almost negligible gap can be detected. Agüero et al. [15] reported the presence of an important gap in PLA–flax fiber composites which was related to poor toughness since this gap does not allow good load transfer from the matrix to the fiber. Figure 4c shows the fracture surface of the BioHDPE-jute composite. As it can be seen, jute fiber is the largest fiber with a diameter comprised between 80 and 150 µm. Despite fiber–polymer interaction being not at its maximum, the presence of PE-*g*-MA contributes to reducing the gap between the polymer matrix and the surrounding fiber. In addition, the pulled-out fibers also decrease with the presence of PE-g-MA. As expected, all three fibers show balanced fiber–polymer interaction which is mainly provided by PE-*g*-MA. The PE chains in PE-*g*-MA tend to interact with BioHDPE polymer chains, while the maleic anhydride groups can interact or even react with the hydroxyl groups in cellulose of all three fibers. The obtained results suggest a relation between the fiber size and the mechanical properties of the corresponding composites. Mazzanti et al. [65] reported the effect of the fiber size on final properties of composites. They concluded that thicker fibers are more effective than thinner ones. 

With regard to the positive effect of PE-*g*-MA, Lima et al. [62] reported improved interface interaction in HDPE-chitosan blends. They also concluded PE-*g*-MA allows better dispersion and prevent aggregate formation. All these phenomena contribute to improve the overall mechanical properties. Other studies have reported the positive effect of acrylic acid as compatibilizer in polypropylene-chitosan composites, which exerts similar effects as PE-*g*-MA [68]. 

### 3.3. Thermal Properties of BioHDPE-Natural Fiber Composites

Figure 5 shows DSC thermograms obtained during the second heating cycle of the BioHDPE composites with different natural fibers. Moreover, Table 3 sums up the main data collected in this analysis. It can be observed that BioHDPE shows its melting peak temperature (T_m_) at 131.1 °C, which is very similar to the value that other studies reported in literature [69]. Furthermore, neat BioHDPE possesses a high crystallinity degree, X_c_ of 69.2%. The addition of PE-*g*-MA does not change in a significant way the melt peak temperature. Quiles-Carrillo et al. [4] have already reported that the incorporation of PE-*g*-MA in BioHDPE/PLA blends does not change the characteristic melting peaks of both immiscible polymers. However, it seems PE-*g*-MA has a clear effect on crystallinity which is reduced by 14% with regard to neat BioHDPE. Abdul Wahab et al. [70] also reported a decrease in crystallinity after the addition of PE-*g*-MA into ternary high density polyethylene/natural rubber/thermoplastic tapioca starch blends. This decrease in crystallinity is related to the distortion induced by maleic anhydride groups.

Referring to the obtained values with the addition of the different natural fibers, the T_m_ of them all remains almost invariable with values comprised between 131 and 132 °C, thus showing that the addition of these reinforcement fibers does not affect T_m_ of the base BioHDPE matrix. Nonetheless, a change in crystallinity can be appreciated. If compared to the crystallinity degree of BioHDPE/PE-*g*-MA blend, the presence of all three fibers leads to a clear increase in crystallinity, reaching X_c_ values of 61.4%, 65.9% and 60.2% for hemp-, flax- and jute-based composites, respectively. These results suggest that fibers could be acting as nucleant, thus favouring formation of more crystals, that counteract the distortion induced by PE-*g*-MA [71]. These results are in total agreement with Pracella et al. [72], who reported the effect of hemp fiber in relation to the increase in the overall crystallinity degree of polypropylene matrix. The increase in crystallization rate in BioHDPE-natural fiber composites could be attributed to the higher nucleation density in the fiber surface.

It must be highlighted that the change in crystallinity of the different fibers is highly linked to their composition (since cellulose can act as a nucleant), but above all to their size. It can be observed how flax fiber, which is the smallest, leads to the highest crystallinity degree (65.9%). Furthermore, jute fiber (the largest one) contributes to the lowest degree of crystallinity for BioHDPE (60.2%). This effect is highly related to the fiber/polymer interactions that restrain the mobility of polymer chains and restrict the crystallization process. This is related to the size and geometry of the fibers as described by Piorkowska et al. [73].

With regard to the thermal stability, Figure 6 shows the thermogravimetric curves of BioHDPE-natural fiber composites and their corresponding DTG curves. Additionally, Table 4 sums up the main degradation parameters of these green composites. The TGA curve of BioHDPE is the one with the highest thermal stability, with a maximum degradation peak temperature (T_deg2_) around 478.9 °C, while the onset temperature (for a weight loss of 5%) is located at 341.6 °C. The TGA blend of BioHDPE and PE-g-MA is similar to that of neat BioHDPE, with a single step degradation process; nevertheless, the characteristic degradation temperatures are moved to lower values. The maximum degradation rate temperature is 469.8 °C, and the onset is close to 331 °C. This behavior has been previously observed by Lima et al. [62], in HDPE/Chitosan composites by using PE-*g*-MA.

It can be observed how the addition of natural fibers significantly reduced the thermal stability of BioHDPE-based composites. Moreover, the degradation process changes significantly from a single step degradation process (neat BioHDPE), to a multi-stage degradation process with the lignocellulosic fibers. In general, lignocellulosic fibers offer lower thermal stability than most polymers. Lignocellulosic fibers degrade in a very similar way due to their composition, as the cellulose-based thermal degradation is very influenced by structure and chemical composition [74]. 

An initial process can be observed at a temperature range of 80–120 °C, which is attributed to the residual water removal. Cellulose and hemicelluloses are the main components of the used fibers. Hemicelluloses are the less thermal stable components with a degradation range from 220–315 °C. Cellulose is more thermally stable and degrades in the 300–400 °C range. With regard to lignin, it degrades along quite a wide temperature range from 250 to above 500 °C [75]. Therefore, as the main components of all three natural fibers are cellulose and hemicellulose, the thermal stability is reduced, compared to neat BioHDPE. As it can be seen in Figure 5, BioHDPE composites with flax and jute, offer better thermal stability than BioHDPE-hemp composites, which can be attributed to the high cellulose and hemicellulose content in hemp fiber [76]. The maximum degradation rate peak temperature for BioHDPE (T_deg2_) in all composites remains close to 480 °C, but a second degradation stage can be clearly observed from DTG curves at lower temperatures. This is mainly related to the cellulose and hemicellulose degradations. The temperature for this degradation process (T_deg1_) is comprised between 350 and 370 °C, which is typical for cellulose degradation [77]. The peak above 500 °C could be attributed to the oxidative decomposition of the carbonized residue [78]. As expected, the most relevant effect is a clear decrease in the onset degradation temperature since cellulose and hemicellulose degrade in the 220–400 °C range, and this is responsible for the initial weight loss during thermal degradation. 

### 3.4. Thermomechanical Properties of PA1010/SFs Composites

Figure 7 shows the dynamic thermal mechanical (DMTA) behavior of neat BioHDPE and the effect of PE-*g*-MA and the three naturals fibers considered in this study. Particularly, Figure 6a shows the evolution of the storage modulus (E′) with temperature. The glass transition temperature (T_g_) of BioHDPE can be clearly identified in Figure 6b, which gathers the plot of the dynamic damping factor (tan δ) as a function of temperature. Despite existing several criteria to define the T_g_ value from DMTA data, one of the most used criteria is the maximum peak of the dynamic damping factor. Thus, the T_g_ of BioHDPE is located at about −120 °C and this transition is related to the amorphous (non-crystalline) polyethylene regions [79]. A second relaxation process appears between 50 and 120 °C, and it is associated to an interlaminar shearing process. This can be divided into two contributions due to the lack of homogeneity of the crystalline regions as reported by Pegoretti et al. [80]. Below its T_g_, neat BioHDPE presents a storage modulus, E’ of 2664 MPa, while it is dramatically reduced at temperatures above 50 °C, with an E’ value of 175 MPa at 75 °C.

The incorporation of fibers in the polymeric matrix directly changes the dynamic mechanical thermal behavior of BioHDPE composites. In particular, a higher storage modulus is obtained in all the temperature range for all three fibers: hemp, flax and jute. Particularly, it is BioHDPE–hemp fiber composite the one with the highest stiffness in all the temperature range. These results are in accordance with those described previously in tensile characterization.

On the other hand, Table 5 sums up the T_g_ values of each composite, altogether with the storage modulus, E’ at different temperatures. With regard to the addition of PE-*g*-MA into BioHDPE, the stiffness is slightly decreased at temperatures below 50 °C, changing from 2664 MPa to 2469 MPa at −145 °C and, from 1151 MPa to 1081 MPa at 0 °C. This behavior is in agreement with the above-mentioned results regarding tensile properties of BioHDPE/PE-*g*-MA blends and FESEM characterization which suggested some plasticization effect of PE-*g*-MA on BioHDPE. On the other hand, it should be noted that the addition of this copolymer does not affect the T_g_ value of the neat BioHDPE. In relation to BioHDPE composites with different natural fibers, the increase in stiffness is noticeable. Particularly it can be observed how at 0 °C, the hemp-based composite offers the highest stiffness with an E’ value of 1525 MPa. In general, the hemp-based composite offers the highest stiffness in comparison with flax- and jute-based composites. Nonetheless, flax and jute fibers increase T_g_ by 6.7 °C and 1.1 °C, respectively, which is directly related to improved polymer–fiber interactions. It seems that flax-based composites offer the best polymer–fiber interaction since the increase in T_g_ is much higher than in hemp- and jute-based composites [71,81]. The addition of hemp fiber slightly decreases the T_g_ value. Castro et al. [79], observed a similar behavior in biobased HDPE composites with curaua fibers. The increase in T_g_ in thermoplastic composites with natural fibers has been observed by Girones et al. [82]. They reported an increase in T_g_ by DMTA in thermoplastic starch (TPS) composites with sisal and hemp fibers, this was attributed to restricted chain mobility due to improved polymer–fiber interactions.

### 3.5. Color Measurement and Visual Appearance 

Loading polymers with natural fillers usually provides a wood-like appearance, which makes them more attractive. Figure 8 shows the visual appearance of BioHDPE composites with hemp, flax and jute fibers. As one can see, by using different natural fibers, it is possible to obtain different brownish colors, directly related to the natural color of fibers. At first sight, hemp fibers provide the brightest brownish color, while flax fibers provide a darker tone with several clear points. Finally, jute-based composite is the darkest one.

In order to analyze the colour in a quantitative way, the colour coordinates (L*, a*, b*) were measured. Table 6 gathers these colour coordinates for neat BioHDPE and its composites with hemp, flax and jute fiber. As it was expected, the darkest sample is the one with the lowest luminance, which corresponds to jute fiber. This gave an L* value of 38.7, while colour coordinates a* and b* were 4.6 and 8.1, respectively. In fact, the incorporation of the fibers provides positive values for the colour coordinates a* and b*, which corroborates that their addition generates a series of materials with reddish-like colours with brown tones. The colour coordinates of the obtained BioHDPE composites are similar to some typical woods such as pine, oak, birch, walnut, or teak, among others, thus opening new potential applications for these materials as wood substitutes as reported by Liminana et al. [83] in polybutylene succinate (PBS) composites with almond shell flour, and other natural woods [84,85]. This fact can be interesting from an aesthetic point of view in packaging elements, such as trays, decoration elements for gardens and interiors or substitute elements of different woods [8].

### 3.6. Water Uptake Characterization

In general, wood plastic composites have the main drawback of containing a high proportion of lignin, cellulose and hemicellulose. These compounds are highly hydrophilic, which is not positive for certain industries and applications, since they are very sensitive to moister and water uptake. As a result, one of the main disadvantages of green composites is their tendency to absorb water. Figure 9 shows the evolution of water absorption of injection-molded pieces during 15 weeks of water immersion.

The neat bioHDPE sample did not exhibit hardly any water absorption, with an asymptotic value of approximately 0.03 wt.%. This observation demonstrates the intrinsic hydrophobic behavior of this material. Addition of PE-*g*-MA leads to a slight increase in water absorption, obtaining an asymptotic value of 0.04 wt.%, since PE-*g*-MA contains highly hydrophilic groups (maleic anhydride), thus contributing to an increased affinity towards water [86]. 

As it was expected, after the incorporation of the fibers, all composites displayed a considerable increase in water absorption, up to values around 1 wt.%. Particularly, a maximum water absorption of 1 wt.%; 0.96 wt.% and, 0.89 wt.% can be observed for the hemp-, flax- and jute-based composites with BioHDPE, respectively. Similar water absorption behavior was reported by Najafi et al. [87] when studying several lignocellulosic fillers in high density polyethylene matrices. Despite all fiber-based composites show increased water absorption, it is hemp-based composite which shows slightly higher water uptake value. 

### 3.7. Fire Behavior of BioHDPE-Natural Fiber Composites

Table 7 gathers the values in relation to the fire behavior of the BioHDPE-natural fiber composites in terms of combustion energy and smoke density values. These values allow a better understanding of how the fire behavior of BioHDPE changes with the incorporation of natural fibers. In relation to the heat release, this value allows to determine the combustion power in MJ/kg of the material. Bio-HDPE shows the worst behavior regarding the heat release since it offers higher combustion power compared to its composites with hemp, flax and jute. During combustion, polyethylene decomposes with very little char formation. It is important to bear in mind that the formed char can isolate the material from oxygen, thus providing a fire retardancy effect. This situation is not observed in polyethylene and hence, it shows the highest heat release [88].

With the addition of PE-*g*-MA, a slight decrease in combustion power is detectable. However, the addition of the fibers does produce a more noticeable change. In particular, a greater reduction in combustion power can be seen with the addition of flax fibers to BioHDPE, followed by jute- and hemp-based composites. This fact is closely related to the structure and composition of this type of fibers which, despite having large amounts of lignin and cellulose, generates a protective barrier. This reduction is due to the fact that natural fibers act as char-forming agents, which can delay the spread of burning and isolate the material from oxygen [89]. Some authors have reported similar behavior for flax fibers. Natural fibers are charred during combustion and also absorb heat from the system, forming a protective layer and porous structure. The charred layer acts as a physical protective barrier against heat transfer, and reduces the heat release in all three BioHDPE composites with hemp, flax and jute [90]. These results are closely related to the values obtained from the opacity test. Although in all cases the density is very high, due to the organic nature of both the polymer and the reinforcement, the material with the best performance is BioHDPE/PE-*g*-MA. When natural fibers are added to BioHDPE matrix, the material’s performance in terms of smoke density worsens. 

On the other hand, the addition of natural fibers significantly reduces the flame starting time (FST) and flame duration (t_inf_). Fibrous materials melt and char, which can be the reason for the flames going out and for the reduction in the flame starting time, as fibers are formed by lignocellulosic compounds, which are more. As in the case of the heat release, the addition of natural fibers had a positive effect on the reduction of t_inf_ due to the formation of a charred barrier layer covering the sample surface and acting as a thermal and gas barrier. From a structural point of view, this layer is formed when the fibers in the composite start to burn, as those fibers have a high carbon content, which is responsible for the formation of a porous char that acts as a barrier. The presence of this layer on the sample surface isolates the material from oxygen and volatiles with a positive effect on fire behavior [91]. Dorez et al. [92], reported this barrier effect in polybutylene succinate (PBS)/natural fiber composites. This protective phenomenon was effective for a fiber content above 10%. This can be attributed to a percolation threshold of the fibers in the composites, thus favoring the cohesion of the residue. 

## 4. Discussion

This work shows that natural short fibers can be effectively used as reinforcing elements in fully biobased high density polyethylene (BioHDPE) parts prepared by conventional industrial processes for thermoplastic materials, as it is the case of injection molding. Regarding mechanical properties, the three fibers drastically increased Young’s modulus, with an increase of more than 100% in relation to the modulus of neat BioHDPE, being hemp fiber the one that provides the highest Young’s modulus (1730 MPa). Thus, verifying the reinforcing effect of the fibers. Tensile strength, hardness and impact strength were also increased for all fibers, which is indicative of very promising results. On the other hand, elongation at break was reduced, which was expected as the addition of embedded fibers into the polymer matrix decreases its continuity and, therefore, the overall cohesion is also reduced. The incorporation of PE-*g*-MA in the composites proved to be a key element in terms of compatibilization, thanks to its dual functionality. Morphology analysis confirmed the results obtained in mechanical tests and demonstrated that the size and shape of the fibers determines the mechanical response. Furthermore, a well-balanced fiber–polymer interaction was observed for all the fibers thanks to the action of PE-*g*-MA, which exerts a compatibilizing effect over the fiber and the polymer. With regard to thermal properties, the melting point of BioHDPE hardly suffered any change, although the three fibers improved thermal stability up to a point. Crystallinity was increased due to the addition of the three natural fibers in comparison with the BioHDPE/PE-*g*-MA blend, being flax fiber the one which provides the highest crystallinity due to being the smallest one. From a thermomechanical point of view, all three fibers increased the storage modulus, especially hemp fiber, which showed a storage modulus of 2944 MPa at −145 °C, confirming the reinforcing effect of the fibers. As expected, hemp, flax and jute fibers increased water absorption of BioHDPE because of their lignocellulosic nature. Finally, in terms of fire behavior, the natural fibers tested here proved to exert a positive effect over the flame duration, as they are capable of forming a charred barrier layer that covers the sample surface and acts as a thermal barrier. In this sense, jute fiber presented the lowest flame duration.

## 5. Conclusions

The results obtained in this work show that it is possible to obtain functional wood plastic composites with high renewable content, based on BioHDPE and natural short fibers. These kind of green composites favor an environmentally friendly approach towards material production as well as they support the circular economy concept. The composites developed in this study have also shown excellent properties at a reduced cost in relation to neat BioHDPE. Improved stiffness and hardness, good compatibility, good thermal stability and improvement in fire behavior should be remarked. All in all, this study demonstrates the viability and potential of hemp, flax and jute fibers to develop new added-value green composites as well as the ability of PE-*g*-MA to exert an excellent compatibility effect over the fibers. This opens a new investigation route through which new compatibilizers could be tested, attempting to even surpass the results obtained in this work.

## Figures and Tables

**Figure 1 polymers-13-01692-f001:**
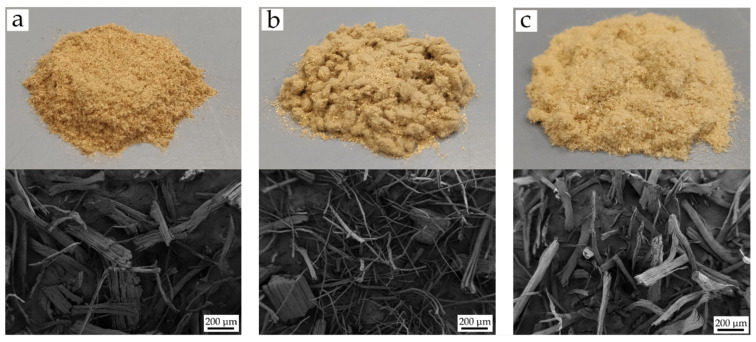
Visual appearance (upper row) and field emission scanning electron microscopy (FESEM) images at 50× (bottom row) of different natural fibers: (**a**) Hemp; (**b**) Flax; (**c**) Jute.

**Figure 2 polymers-13-01692-f002:**
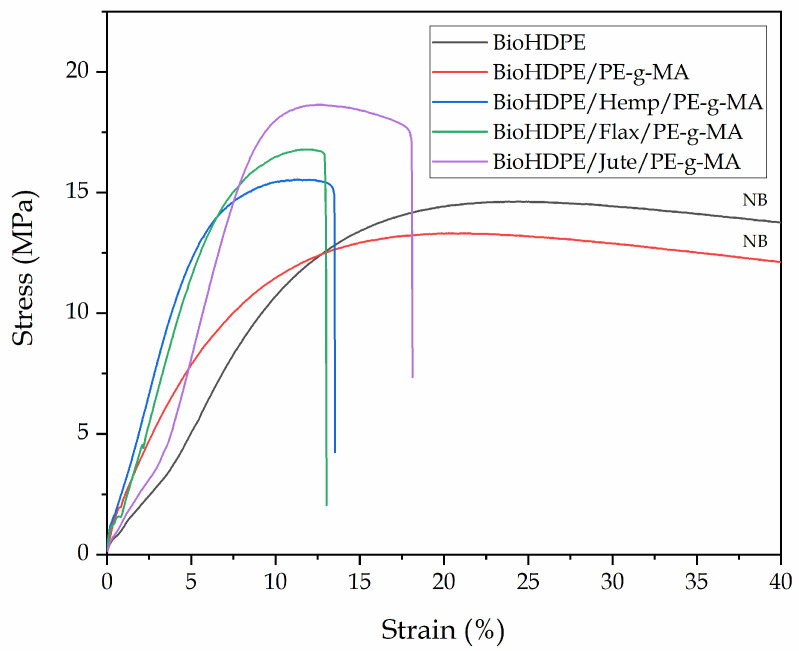
Stress strain curves for all the BioHDPE/PE-g-MA composites reinforced with natural fibers.

**Figure 3 polymers-13-01692-f003:**
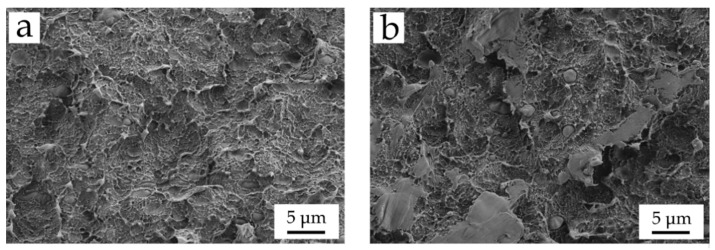
Field emission scanning electron microscopy (FESEM) images at 2000× of the fracture surfaces of (**a**) neat BioHDPE, and (**b**) BioHDPE/PE-g-MA blend.

**Figure 4 polymers-13-01692-f004:**
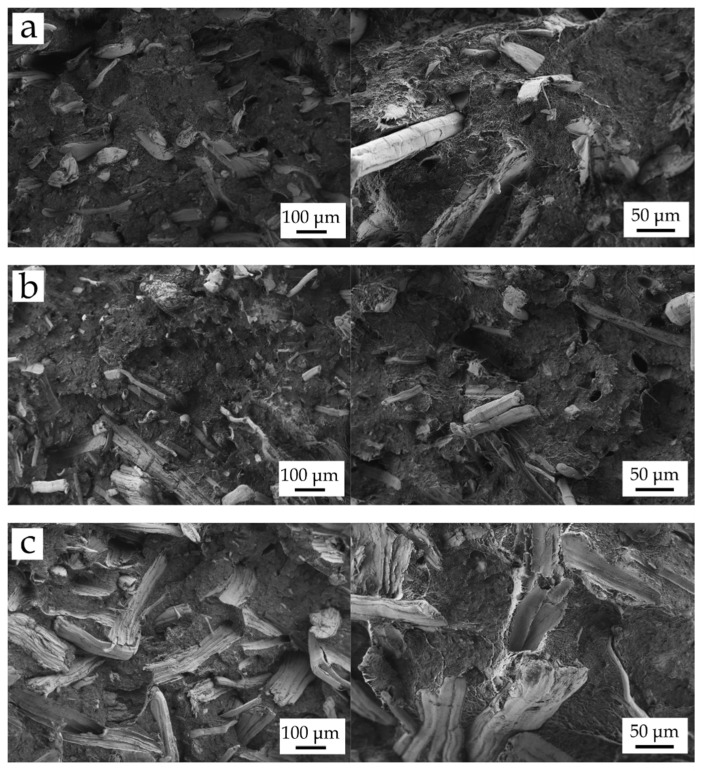
Field emission scanning electron microscopy (FESEM) images at 100× (left) and 250× (right) of the fracture surfaces of the different BioHDPE composites: (**a**) BioHDPE/Hemp/PE-*g*-MA; (**b**) BioHDPE/Flax/PE-*g*-MA and (**c**) BioHDPE/Jute/PE-*g*-MA.

**Figure 5 polymers-13-01692-f005:**
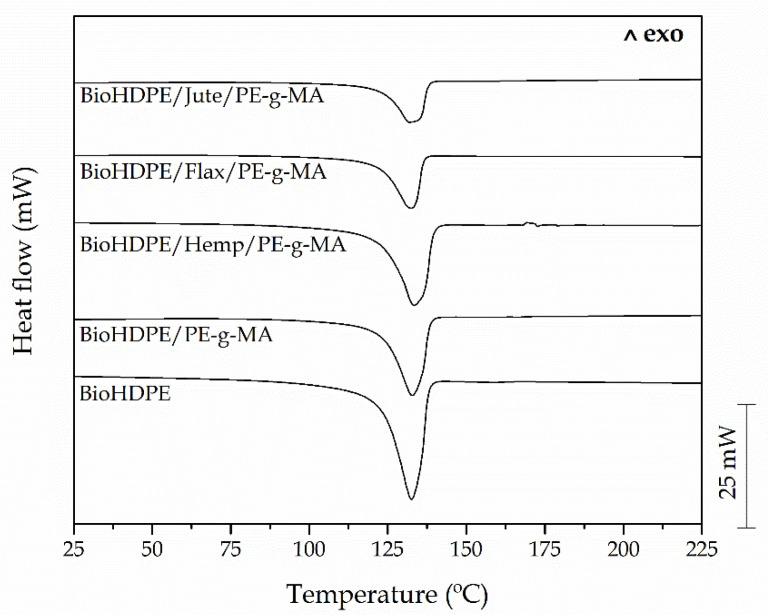
Differential scanning calorimetry (DSC) thermograms of BioHDPE composites with different natural fibers.

**Figure 6 polymers-13-01692-f006:**
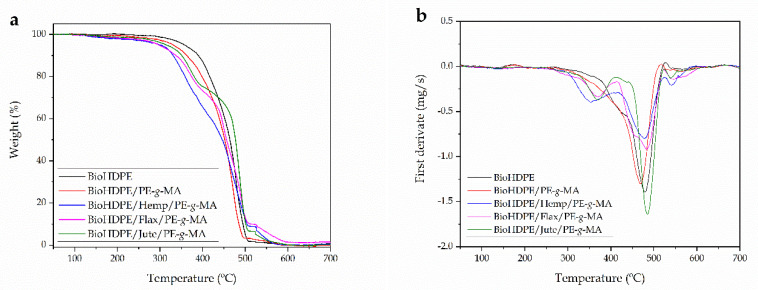
Thermal degradation of BioHDPE composites with different natural fibers, (**a**) thermogravimetric (TGA) curves and (**b**) first derivative (DTG) curves.

**Figure 7 polymers-13-01692-f007:**
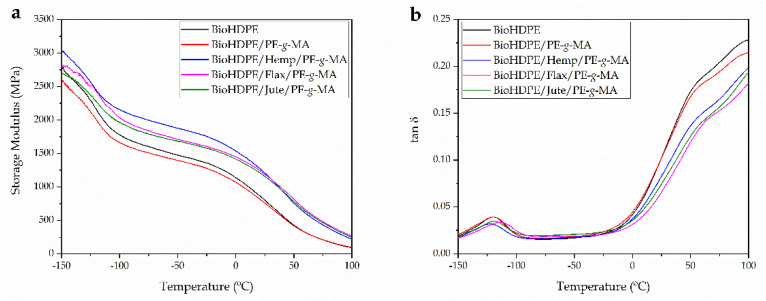
Dynamic mechanical thermal behavior (DMTA) of Bio-HDPE with different natural fibers as a function of temperature, (**a**) storage modulus (E’) and (**b**) dynamic damping factor (tan δ).

**Figure 8 polymers-13-01692-f008:**
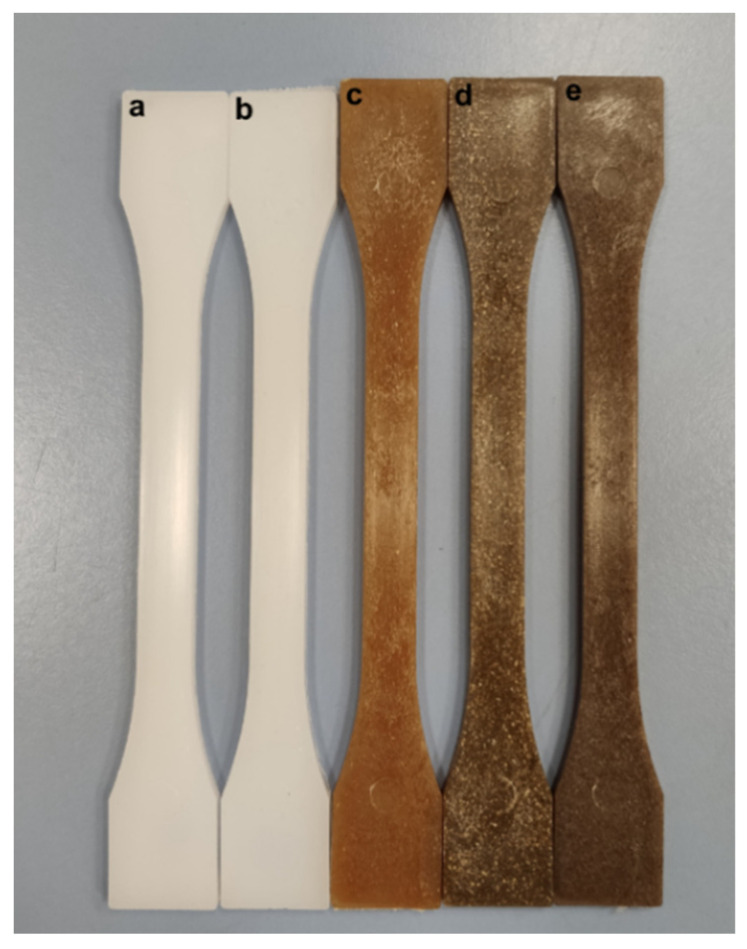
Visual appearance of the samples: (**a**) BioHDPE; (**b**) BioHDPE/PE-g-MA; (**c**) BioHDPE/Hemp/PE-g-MA; (**d**) BioHDPE/Flax/PE-g-MA; (**e**) BioHDPE/Jute/PE-g-MA.

**Figure 9 polymers-13-01692-f009:**
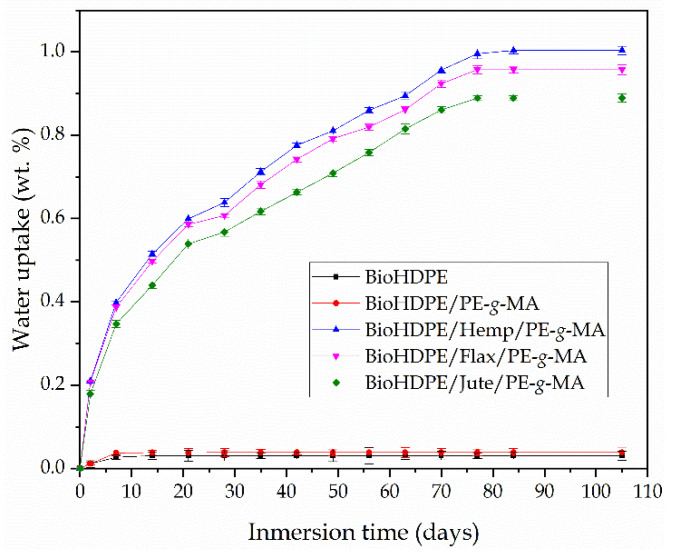
Water uptake of bioHDPE composites with different natural fibers.

**Table 1 polymers-13-01692-t001:** Summary of compositions according to the volume fractions (υ) of BioHDPE, PE-g-MA and different natural fibers.

Code	BioHDPE υ_Bio-HDPE_	Hemp υ_Hemp_	Flax υ_Flax_	Jute υ_jute_	PE-g-MA υ_PE-g-MA_
BioHDPE	1.000	-	-	-	-
BioHDPE/PE-*g*-MA	0.980	-	-	-	0.020
BioHDPE/Hemp/PE-*g*-MA	0.843	0.135	-	-	0.022
BioHDPE/Flax/PE-*g*-MA	0.840	-	0.138	-	0.022
BioHDPE/Jute/PE-*g*-MA	0.839	-	-	0.139	0.022

**Table 2 polymers-13-01692-t002:** Summary of mechanical properties of the injection-molded parts of BioHDPE-natural fiber composites in terms of: tensile modulus (E), maximum tensile strength (σ_max_), elongation at break (ε_b_), Shore D hardness, and impact strength.

Code	E (GPa)	σ_max_ (MPa)	ε_b_ (%)	Shore D Hardness	Impact Strength (kJ/m^2^)
BioHDPE	0.790 ± 0.008	14.5 ± 0.8	NB	56.0 ± 0.7	2.7 ± 0.2
BioHDPE/PE-*g*-MA	0.791 ± 0.014	13.3 ± 0.6	NB	55.8 ± 0.8	2.8 ± 0.3
BioHDPE/Hemp/PE-*g*-MA	1.730 ± 0.014	15.5 ± 1.4	3.4 ± 0.4	59.4 ± 1.1	3.9 ± 0.2
BioHDPE/Flax/PE-*g*-MA	1.630 ± 0.038	16.7 ± 1.1	2.8 ± 0.7	61.6 ± 0.5	3.0 ± 0.1
BioHDPE/Jute/PE-*g*-MA	1.675 ± 0.012	18.6 ± 0.3	4.4 ± 0.9	60.2 ± 1.2	3.7 ± 0.2

NB—Not Break.

**Table 3 polymers-13-01692-t003:** Main thermal parameters of the injection-molded parts of Bio-HDPE with different natural fibers in terms of: melting temperature (T_m_), normalized melting enthalpy (∆H_m_), and degree of crystallinity (χ_c_).

Code	T_m_ (°C)	∆H_m_ (J/g)	χ_c_ (%)
BioHDPE	131.1 ± 1.5	202.9 ± 1.6	69.2 ± 1.5
BioHDPE/PE-*g*-MA	132.1 ± 1.1	161.8 ± 1.2	55.2 ± 0.9
BioHDPE/Hemp/PE-*g*-MA	132.4 ± 0.9	143.9 ± 1.1	61.4 ± 1.1
BioHDPE/Flax/PE-*g*-MA	131.9 ± 1.1	154.4 ± 1.3	65.9 ± 1.2
BioHDPE/Jute/PE-*g*-MA	131.6 ± 0.8	141.1 ± 0.9	60.2 ± 0.8

**Table 4 polymers-13-01692-t004:** Main thermal degradation parameters of the samples of Bio-HDPE with different natural fibers in terms of: temperature at mass loss of 5% (T_5%_), degradation temperature (T_deg_), and residual mass at 700 °C.

Parts	T_5%_ (°C)	T_deg1_ (°C)	T_deg2_ (°C)	Residual Weight (%)
BioHDPE	341.6 ± 1.5	-	478.9 ± 2.6	0.4 ± 0.2
BioHDPE/PE-*g*-MA	331.3 ± 1.2	-	469.8 ± 1.7	0.1 ± 0.1
BioHDPE/Hemp/PE-*g*-MA	300.3 ± 1.1	352.8 ± 1.0	478.8 ± 1.5	1.6 ± 0.3
BioHDPE/Flax/PE-*g*-MA	305.2 ± 1.1	370.0 ± 1.4	482.9 ± 0.9	2.2 ± 0.4
BioHDPE/Jute/PE-*g*-MA	314.1 ± 0.9	369.9 ± 1.1	481.6 ± 1.2	1.5 ± 0.3

**Table 5 polymers-13-01692-t005:** Main thermomechanical properties of Bio-HDPE composites with different natural fibers obtained by dynamic mechanical thermal analysis (DMTA).

Parts	E’ (MPa) at −145 °C	E’ (MPa) at 0 °C	E’ (MPa) at 75 °C	T_g_ (°C)
BioHDPE	2664 ± 18	1151 ± 10	210 ± 5	−120.1 ± 0.8
BioHDPE/PE-*g*-MA	2469 ± 19	1081 ± 9	210 ± 3	−120.0 ± 0.9
BioHDPE/Hemp/PE-*g*-MA	2944 ± 29	1525 ± 14	430 ± 8	−122.4 ± 0.9
BioHDPE/Flax/PE-*g*-MA	2781 ± 35	1429 ± 12	485 ± 9	−113.3 ± 1.1
BioHDPE/Jute/PE-*g*-MA	2648 ± 28	1400 ± 14	460 ± 10	−118.9 ± 1.0

**Table 6 polymers-13-01692-t006:** Luminance and color coordinates (L*, a*, b*) of BioHDPE composites with different natural fibers.

Code	L*	a*	b*
BioHDPE	72.5 ± 0.5	−2.1 ± 0.1	−5.5 ± 0.1
BioHDPE/PE-*g*-MA	71.8 ± 0.7	−2.3 ± 0.1	−5.1 ± 0.1
BioHDPE/Hemp/PE-*g*-MA	44.9 ± 0.4	7.3 ± 0.1	17.0 ± 0.2
BioHDPE/Flax/PE-*g*-MA	40.7 ± 0.4	3.6 ± 0.1	10.4 ± 0.1
BioHDPE/Jute/PE-*g*-MA	38.7 ± 0.2	4.6 ± 0.1	8.1 ± 0.1

**Table 7 polymers-13-01692-t007:** Summary of thermal parameters obtained with the calorimetry and opacity test on BioHDPE composites with different natural fibers.

Code	Heat Release (MJ/kg)	Ds_max_	FST(s)	t_inf_ (s)
BioHDPE	46.5 ± 3.1	640.3 ± 5.9	159 ± 7	>600 ± 12
BioHDPE/PE-*g*-MA	44.8 ± 2.5	580.9 ± 7.5	210 ± 9	>600 ± 10
BioHDPE/Hemp/PE-*g*-MA	42.5 ± 1.9	864.8 ± 9.8	56 ± 3	303 ± 10
BioHDPE/Flax/PE-*g*-MA	40.6 ± 2.1	856.2 ± 14.9	53 ± 4	329 ± 12
BioHDPE/Jute/PE-*g*-MA	41.1 ± 2.3	792.8 ± 12.6	46 ± 3	294 ± 5

## Data Availability

The data presented in this study are available on request from the corresponding author.

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
