# Peer review of "Development and Characterization of Environmentally Friendly Wood Plastic Composites from Biobased Polyethylene and Short Natural Fibers Processed by Injection Moulding"

_polymers, 2021, doi:10.3390/polym13111692_

Round 1
Reviewer 1 Report
Dear authors,
The review paper related to the development and characterization of environmentally friendly wood-plastic composites from biobased polyethylene and short natural fibers processed by injection molding has been reviewed. As one of the selected reviewers, I read your manuscript carefully. While your research has addressed an important subject, I observed some issues that must be addressed in a newer version.
The aim of the study at the end of the introduction requires clarification. Why did you study different fiber/HDPE ratios? How were they chosen? Why do you want to develop wood-plastic composites?
You should also emphasize the novelty of the study: HDPE-natural fiber composites have been previously extensively studied; what's the novelty here?
Please check the value of elongation at break of jute fiber, i.e., 13%, on the second last line of page 3. Are you sure about this value? The value is ten times higher than the two other fibers discussed in this study.
Provide the instrument manufacturer, city, and country for all materials and equipment.
The authors have missed some important wood-plastic composites, i.e., Abutilon reinforced with PLA, that fully fit the context of short natural fibers processed by injection molding wood-plastic composites. The authors might read about rheological and dynamic mechanical properties of Abutilon natural straw and polylactic acid biocomposites as well as characterization of natural composites fabricated from Abutilon-fiber-reinforced poly (lactic acid).
Can the authors please justify the change in flame retardance with respect to the structure-property relationship?
The authors might refer to such similar less known fibers that have a high potential for application in the development of wood-plastic composites.
The conclusion section is missing. Please make some concrete conclusions. I think the conclusion section should be written briefly and concretely to address the key findings.
What are the prospects? Add a separate section or briefly add few sentences in the conclusion section? Also, no research limitation is explained aligned with the research problem. The research limitations describe what dimensions of the problem are excluded by you and your study's boundaries.
I could not access the supplementary materials. If this manuscript has any Supplementary Materials, make sure to mention them in the text or delete that section.
I look forward to receiving the revision and reviewing the newer version.
Best of luck.
Reviewer 2 Report
Presented manuscript is a good work that can be accepted after minor revisions.
- I suggest to perform FTIR and TGA analysis for bio-based fillers. FTIR spectra of bioPE and PE-g-MA should be also presented and analysed.
- The wt% of MA in PE-g-MA should be indicated
- Please, present examplary stress-strain curves
Reviewer 3 Report
The authors presented intensive range of experiments for a potential environmentally friendly plastic composites produced with natural fibres and bio-based high density polyethylene via injection moulding. This work and the results of potentially sustainable natural fibre reinforced bio-based plastic composites, timely important field, would be beneficial for readers. But the submitted manuscript is a long manuscript that shows so much information that includes some weaknesses and sadly, there is no clear message for a reader. I do understand the importance of work however putting intensive amount of experimental results work should not confuse readers. Please reconsider what the work tells for a reader and change the manuscript according to that. Also, I strongly disagree with the use of "novel" in discussion section line 605 - and end of introduction should change there is no "high environmental efficiency materials" terminology is confusing and there is no proof for that or a valid reference for that. Therefore, the current manuscript cannot be accepted, a major revision is needed. Please also satisfy following comments and suggestions:
-Line 84-86 two sentences have a conflict and first sentence is unclear, what composition?
-Line 92 why flax has its scientific name the others do not have? please either remove it or put name for other fibres too.
-Line 107-112, Confusing and there is no meaning for these sentences. Please re-rewrite.
-Line 122 "high environmental efficiency materials"? Please introduce the aim of work without vague words.
-Introduction- Please reference recently published similar works about aligned short natural fibre composites and circular economy which will interest potential readers (Materials 2021, 14(8), 1885; https://doi.org/10.3390/ma14081885, Sustainability 2021, 13(3), 1160; https://doi.org/10.3390/su13031160)
-Line 144-149 terminology of coarseness is wrong. It's just a diameter or dimension. Instead of these sentences, a table would be better than a text introduction for the fibre properties. (Figure 1 introduction is wrong, it doesn't only show SEM images! Please change the caption too.)
-Line 157 "the components were pre-mixed in a zipper bag"? Why should a reader know this? Please remove.
-Table 1 it would be better to show also approximate volume fractions since it makes more sense for a reader who is in fibre reinforced polymer composite field.
-Section 3 please make symbols for tensile strength and elongation at break consistent in the section.
-Table 2 Elastic Modulus must be given in terms of GPa. Representative stress-strain curves must be given.
-Line 274 "Authors such as"?! Please remove.
-Figure 8 I found the water uptake values suspiciously low, and SD values looks same for all of data points. Please re-check and share the data in your responses report.
-Line 553 "increased sensitiveness"? I don't understand this at all!
-Line 559-562 Is this information necessary? There looks no statistically significant difference to make a statement. It would be better to do statistical analysis than vague sentences.
Round 2
Reviewer 1 Report
The authors have improved the manuscript as per the reviewer's guidance; therefore, the manuscript might be considered for publication in its present form.
Author Response
Dear Reviewer,
We deeply appreciate your rapid response and we are really thankful for your positive veredict. We would like to thank you again for your previous suggestions to make our manuscript better and improve our work.
Best regards.
Jaume
Reviewer 3 Report
The authors answered and satisfied most of the comments, the revised manuscript can be accepted for a publication in its present form after the minor points below were applied.
-Please mention Young's Modulus always in GPa (not only in the table also in the text)
-Please check the studies in fibre reinforced polymer composite field to see how volume fractions are represented properly (for Table 1) - also check the Table 1 for typos (caption should include the information about volume fractions and v/v is not the correct way of representing them)
